# A Data-Driven and Distributed Approach to Sparse Signal Representation and Recovery

**Ali Mousavi**
Google AI
alimous@google.com

**Gautam Dasarathy**
Arizona State University
gautamd@asu.edu

**Richard G. Baraniuk**
Rice University
richb@rice.edu

## Abstract

In this paper, we focus on two challenges which offset the promise of sparse signal representation, sensing, and recovery. First, real-world signals can seldom be described as perfectly sparse vectors in a known basis, and traditionally used random measurement schemes are seldom optimal for sensing them. Second, existing signal recovery algorithms are usually not fast enough to make them applicable to real-time problems. In this paper, we address these two challenges by presenting a novel framework based on deep learning. For the first challenge, we cast the problem of finding informative measurements by using a maximum likelihood (ML) formulation and show how we can build a data-driven dimensionality reduction protocol for sensing signals using convolutional architectures. For the second challenge, we discuss and analyze a novel parallelization scheme and show it significantly speeds-up the signal recovery process. We demonstrate the significant improvement our method obtains over competing methods through a series of experiments.

## 1 Introduction

High-dimensional inverse problems and low-dimensional embeddings play a key role in a wide range of applications in machine learning and signal processing. In inverse problems, the goal is to recover a signal $X \in \mathbb{R}^N$ from a set of measurements $Y = \Phi(X) \in \mathbb{R}^M$, where $\Phi$ is a linear or non-linear sensing operator. A special case of this problem is compressive sensing (CS) which is a technique for efficiently acquiring and reconstructing a sparse signal (Donoho, 2006; Candès et al., 2006; Baraniuk, 2007). In CS $\Phi \in \mathbb{R}^{M \times N}$ ($M \ll N$) is typically chosen to be a random matrix resulting in a random low-dimensional embedding of signals. In addition, $X$ is assumed to be sparse in some basis $\Gamma$, i.e., $X = \Gamma S$, where $\|S\|_0 = K \ll N$.

While sparse signal representation and recovery have made significant real-world impact in various fields over the past decade (Siemens, 2017), arguably their promise has not been fully realized. The reasons for this can be boiled down to two major challenges: First, real-world signals are only approximately sparse and hence, random/universal sensing matrices are sub-optimal measurement operators. Second, many existing recovery algorithms, while provably statistically optimal, are slow to converge. In this paper, we propose a new framework that simultaneously takes on both these challenges.

To tackle the first challenge, we formulate the learning of the dimensionality reduction (i.e., signal sensing operator) as a likelihood maximization problem; this problem is related to the Infomax principle (Linsker, 1989) asymptotically. We then show that the simultaneous learning of dimensionality reduction and reconstruction function using this formulation gives a lower-bound of the objective functions that needs to be optimized in learning the dimensionality reduction. This is similar in spirit to what Vincent et al. show for denoising autoencoders in the non-asymptotic setting (Vincent et al., 2010). Furthermore, we show that our framework can learn dimensionality reductions that preserve specific geometric properties. As an example, we demonstrate how we can construct a data-driven near-isometric low-dimensional embedding that outperforms competing embedding algorithms like NuMax (Hegde et al., 2015). Towards tackling the second challenge, we introduce a parallelization (i.e., rearrangement) scheme that significantly speeds up the signal sensing and recovery process. We show that our framework can outperform state-of-the-art signal recovery methods such as DAMP

(Metzler et al., 2016) and LDAMP (Metzler et al., 2017) both in terms of inference performance and computational efficiency.

We now present a brief overview of prior work on embedding and signal recovery. Beyond random matrices, there are other frameworks developed for deterministic construction of linear (or nonlinear) near-isometric embeddings (Hegde et al., 2015; Grant et al., 2013; Bah et al., 2013; Tenenbaum et al., 2000; Weinberger & Saul, 2006; Broomhead & Kirby, 2001; 2005; Verma, 2013; Shaw & Jebara, 2007). However, these approaches are either computationally expensive, not generalizable to out-of-sample data points, or perform poorly in terms of isometry. Our framework for low-dimensional embedding shows outstanding performance on all these aspects with real datasets. Algorithms for recovering signals from undersampled measurements can be categorized based on how they exploit prior knowledge of a signal distribution. They could use hand-designed priors (Candès & Tao, 2005; Donoho et al., 2009; Daubechies et al., 2004; Needell & Tropp, 2009), combine hand-designed algorithms with data-driven priors (Metzler et al., 2017; Borgerding & Schniter, 2016; Kamilov & Mansour, 2016; Chang et al., 2017; Gregor & LeCun, 2010), or take a purely data-driven approach (Mousavi et al., 2015; Kulkarni et al., 2016; Mousavi & Baraniuk, 2017; Yao et al., 2017). As one moves from hand-designed approaches to data-driven approaches, models lose simplicity and generalizability while becoming more complex and more specifically tailored for a particular class of signals of interest.

Our framework for sensing and recovering sparse signals can be considered as a variant of a convolutional autoencoder where the encoder is linear and the decoder is nonlinear and specifically designed for CS application. In addition, both encoder and decoder contain rearrangement layers which significantly speed up the signal sensing and recovery process, as we discuss later. Convolutional autoencoder has been previously used for image compression (Jiang et al., 2017); however, our work is mainly focused on the CS application rather than image compression. In CS, measurements are abstract and linear whereas in the image compression application measurements are a compressed version of the original image and are nonlinear. Authors in Jiang et al. (2017) have used bicubic interpolation for upscaling images; however, our framework uses a data-driven approach for upscaling measurements. Finally, unlike the image compression application, when we deploy our framework for CS and during the test phase, we do not have high-resolution images beforehand. In addition to image compression, there have been previous works (Shi et al., 2017; Kulkarni et al., 2016) to jointly learn the signal sensing and reconstruction algorithm in CS using convolutional networks. However, the problem with these works is that they divide images into small blocks and recover each block separately. This blocky reconstruction approach is unrealistic in applications such as medical imaging (e.g. MRI) where the measurement operator is a Fourier matrix and hence we cannot have blocky reconstruction. Since both papers are designed for block-based recovery whereas our method senses/recovers images without subdivision, we have not compared against them. Note that our method could be easily modified to learn near-optimal frequency bands for medical imaging applications. In addition, Shi et al. (2017) and Kulkarni et al. (2016) use an extra denoiser (e.g. BM3D, DCN) for denoising the final reconstruction while our framework does not use any extra denoiser and yet outperforms state-of-the-art results as we show later.

Beside using convolutional autoencoders, authors in Wu et al. (2018) have introduced the *sparse recovery autoencoder* (SRA). In SRA, the encoder is a fully-connected layer while in this work, the encoder has a convolutional structure and is basically a circulant matrix. For large-scale problems, learning a fully-connected layer (as in the SRA encoder) is significantly more challenging than learning convolutional layers (as in our encoder). In SRA, the decoder is a $T$-step projected subgradient. However, in this work, the decoder is several convolutional layers plus a rearranging layer. It should also be noted that the optimization in SRA is solely over the measurement matrix and $T$ (which is the number of layers in the decoder) scalar values. However, here, the optimization is performed over convolution weights and biases that we have across different layers of our network.

## 2 ARCHITECTURE

In this section, we describe our framework for sparse signal representation and recovery and demonstrate how we can learn (near-)optimal projections and speed up signal recovery using parallelization along with convolutional layers. We call our framework by *DeepSSRR*, which stands for *Deep Sparse Signal Representation and Recovery*.

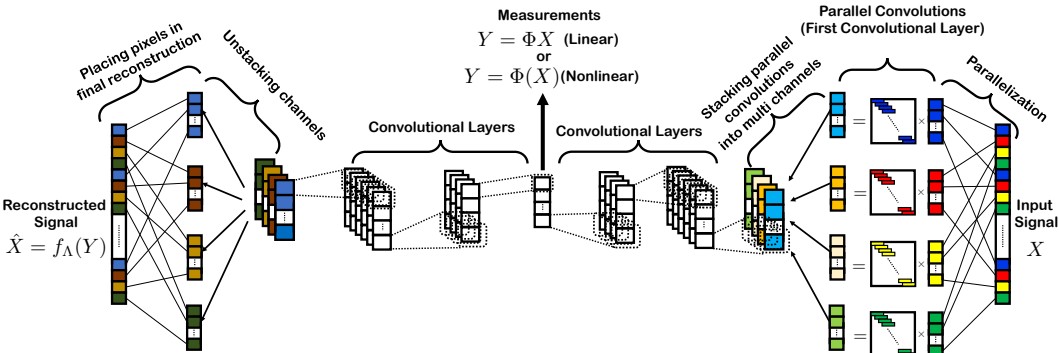

Figure 1: DeepSSRR uses convolutional layers to learn a transformation from signals to undersampled measurements and an inverse transformation from undersampled measurements to signals. Note that operations are performed right to left.

## 2.1 SENSING AND RECOVERY

DeepSSRR consists of two parts: A linear dimensionality reduction $\Phi : \mathbb{R}^N \to \mathbb{R}^M$ for taking undersampled measurements and a nonlinear inverse mapping $f_\Lambda(.) : \mathbb{R}^M \to \mathbb{R}^N$ for recovering signals from their undersampled measurements. We learn both $\Phi$ and $f_\Lambda(.)$ from training data. DeepSSRR (Figure 1) is based primarily on deep convolutional networks (DCN) as this gives us two advantages: (a) sparse connectivity of neurons, and (b) having shared weights which increases learning speed compared to fully-connected networks. Therefore, we impose a convolutional network architecture on both $\Phi$ and $f_\Lambda(.)$ while learning them. Please note that we assume that measurements are linear; however, it is easy to extend DeepSSRR to adopt nonlinear measurements, i.e., allowing for $\Phi(.)$ to be nonlinear by adding nonlinear units to convolutional layers. Given that the intervening layers are linear, one might argue that one convolutional layer (i.e., a single circulant matrix) is enough since we can merge kernel matrices into a single matrix. However, we consider a multi-layer architecture for learning $\Phi$ for two reasons. First, computationally it is cheaper to have separate and smaller kernels and second, it makes the implementation suitable for adding the aforementioned nonlinearities.

We previously mentioned that in order to speed up the sensing and recovery process, we add a parallelization scheme in learning both $\Phi$ and $f_\Lambda(.)$ that we describe in the following. Our original sensing model was $Y = \Phi X$ where $X \in \mathbb{R}^N$ and $Y \in \mathbb{R}^M$. Assume that the undersampling ratio, i.e., $\frac{M}{N}$ is equal to $\frac{1}{r}$. The left vector-matrix multiplication in Figure 2(a) denotes a convolution of zero-padded input signal with size $N' = rM' = r(M + q - 1)$, filter size $rq$, stride (i.e., filter shift at every step) of size $r$, and output size of $M$. If we denote the input signal by $X^{(\text{in})}$ and output by $X^{(\text{out})}$ and filter by $W$ we can write

$$X_j^{(\text{out})} = \sum_{i=1}^{rq} W_i X_{(j-1)r+i}^{(\text{in})} = \sum_{z=0}^{q-1} \left( \sum_{\substack{i=1 \\ i \stackrel{q}{\equiv} z}}^{rq} W_i X_{(j-1)r+i}^{(\text{in})} \right). \tag{1}$$

If we concatenate the sub-filters and sub-signals denoted in orange in the left vector-matrix multiplication of Figure 2(a), we derive a new vector-matrix multiplication shown on the right side of Figure 2(a). There the input size is $M' = (M + q - 1)$, filter size is $q$, stride size is 1, and output size is $M$. Equation (1) states that the left convolution in Figure 2(a) can be written as the summation of $r$ separate and parallel convolutions shown on the right side. Much like in the sensing part (i.e., learning $\Phi$), as shown in Figure 2(b), a large strided deconvolution can be chopped into several parallel smaller deconvolutions for the recovery part (i.e., learning $f_\Lambda(.)$). Because of these parallelizations, the computational complexity of calculating the outputs of layers in DeepSSRR is $\mathcal{O}(M)$ which is much less than the one for typical iterative and unrolled algorithms $\mathcal{O}(MN)$ (e.g. DAMP and LDAMP (Metzler et al., 2016; 2017)) or previous recovery algorithms based on deep learning $\mathcal{O}(N)$ (e.g. DeepInverse (Mousavi & Baraniuk, 2017)).

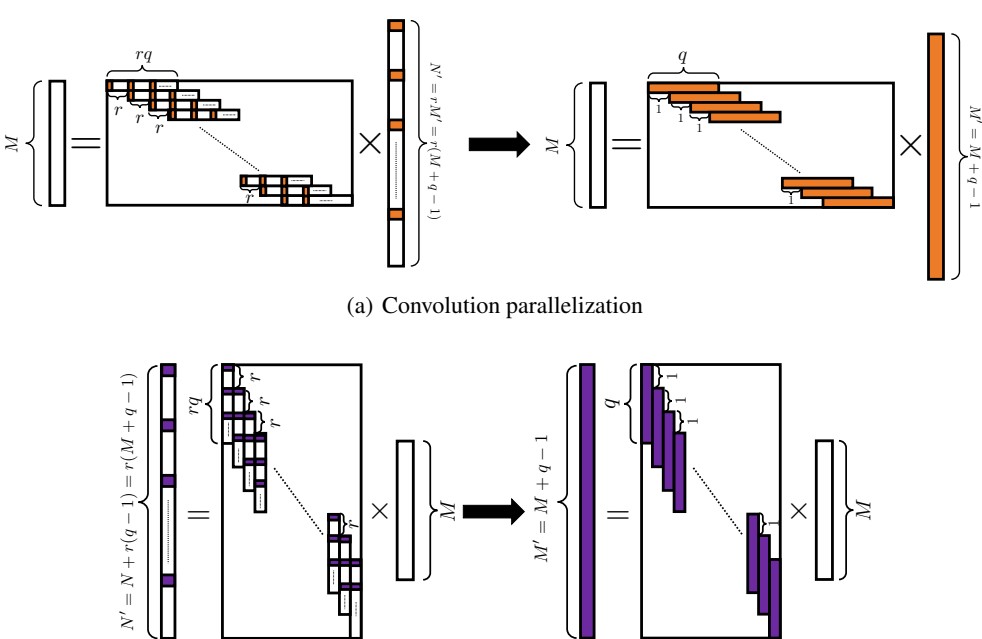

(a) Convolution parallelization

(b) Deconvolution parallelization

Figure 2: Graphical interpretation of convolution parallelization in sensing (right side of Figure 1) and deconvolution parallelization in recovery (left side of Figure 1): Converting a strided convolution (deconvolution) into the summation of several parallel convolutions (deconvolutions).

---

**Algorithm 1** Learning a Near-Isometric Embedding

---

**Input:** Training Dataset $\mathcal{D}$, Number of Epochs $n_{\text{epochs}}$, Network Parameters $\Omega_e$
**Output:** A near-isometric embedding $\Phi : \mathbb{R}^N \to \mathbb{R}^M$
**for** $i = 1$ **to** $n_{\text{epochs}}$ **do**
  - generate a randomly permuted training set $\to \mathcal{P}(\mathcal{D})$
  **for** every batch $\mathcal{B}_j \in \mathcal{P}(\mathcal{D})$ **do**
    - compute embedding $\Phi(X)$ for every $\mathbf{x} \in \mathcal{B}_j$
    - compute the loss function corresponding to $\mathcal{B}_j$
     as the maximum deviation from isometry
$$\mathcal{L}_{\mathcal{B}_j} = \max_{l,k} \left( \frac{\|\Phi(X_l) - \Phi(X_k)\|_2}{\|\mathbf{x}_l - \mathbf{x}_k\|_2} - 1 \right)^2$$
  **end for**
  - compute the aggregated loss function
    $\mathcal{L}(\Omega_e) = \text{avg}_j(\mathcal{L}_{\mathcal{B}_j})$
  - use ADAM optimizer and $\mathcal{L}(\Omega_e)$ to update $\Omega_e$
**end for**

---

As DeepSSRR architecture is shown in Figure 1, For learning $\Phi$, we first divide the input signal (of size $N$) into $r$ ($r = \frac{N}{M}$) sub-signals (of size $M$) such that all the congruent entries (modulo $r$) are in the same sub-signal. Then we run parallel convolutions on $r$ sub-signals and stack the outputs (of size $M$), deriving a tensor of length $M$ and depth $r$. Through several convolutional layers, we turn this tensor into a vector of size $M$ which is the measurements vector $Y$ and this completes construction of $\Phi$. Similarly and for learning $f_\Lambda(.)$, through several convolutional layers, we turn vector $Y$ into a tensor of length $M$ and depth $r$. We then unstack channels similar to the sub-pixel layer architecture (Shi et al., 2016) and derive the final reconstruction $\widehat{X} = f_\Lambda(Y) = f_\Lambda(\Phi X)$. We use MSE as a loss function and ADAM (Kingma & Ba, 2014) to learn the convolution kernels and biases.

**Theoretical Insights.** Notice that CS is the problem of recovering $X \in \mathbb{R}^N$ from $Y = \Phi X \in \mathbb{R}^M$ where $M \ll N$. Therefore, an important question is how does one design $\Phi$? Conventional CS is based on random projections of a signal which means that $\Phi$ is a random matrix in conventional CS. However, since signals are usually structured, random projections are not optimal for successfully recovering the corresponding signals. In many applications (e.g. medical imaging), we know a lot about the signals we are acquiring. Hence, given a large-scale dataset of the same type of signals of interest, we can learn (near-)optimal measurement matrices. As in the usual CS paradigm, if we assume that the measurement matrix $\Phi$ is fixed, each $Y_i$ $(1 \leq i \leq M)$ is a linear combination of $X_j$s $(1 \leq j \leq N)$. We assume the training set $\mathcal{D}_{\text{train}} = \{(X^{(1)}, Y^{(1)}), (X^{(2)}, Y^{(2)}), \ldots, (X^{(\ell)}, Y^{(\ell)})\}$ consists of $\ell$ pairs of signals and their corresponding measurements. Accordingly, we define the optimal measurement operator $\widehat{\Phi}$ as the one which maximizes the probability of training data given the undersampled projections, $\widehat{\Phi} = \arg \max_{\Phi} \prod_{i=1}^{\ell} \mathbb{P}(X^{(i)}|Y^{(i)})$. According to the law of large numbers, notice that we can write

$$\widehat{\Phi} = \arg \max_{\Phi} \lim_{\ell \to \infty} \left( \prod_{i=1}^{\ell} \mathbb{P}(X^{(i)}|Y^{(i)}) \right)^{\frac{1}{\ell}} \tag{2}$$

$$= \arg \max_{\Phi} e^{\mathbb{E}[\log(\mathbb{P}(X|Y))]}$$

$$\overset{(a)}{=} \arg \max_{\Phi} \mathbb{I}(X, Y),$$

where in (a) $\mathbb{I}$ denotes the mutual information, and the equality follows since the Shannon entropy $\mathbb{H}(X)$ is constant for every $\Phi$. According to (2), in the asymptotic setting, the measurement matrix which maximizes the probability of training data given its measurements, maximizes the mutual information between the input signal and undersampled measurements as well. Equation (2) is the same as *infomax* principle first introduced in Linsker (1989).

Now, suppose that we have a function $f(.) : \mathbb{R}^M \to \mathbb{R}^N$ parametrized by $\Lambda$ that receives undersampled measurements $Y^{(i)}$ $(1 \leq i \leq \ell)$ and reconstructs input signals as $\widehat{X}^{(i)} = f_\Lambda(Y^{(i)})$ $(1 \leq i \leq \ell)$. We define the best reconstruction as the one which generates training data with the highest probability. In other words, we define

$$\widehat{\Phi}, \widehat{\Lambda} = \arg \max_{\Phi, \Lambda} \prod_{i=1}^{\ell} \mathbb{P}(X^{(i)}|\widehat{X}^{(i)}).$$

Therefore, in the asymptotic setting and similar to (2) we can write

$$\widehat{\Phi}, \widehat{\Lambda} = \arg \max_{\Phi, \Lambda} \lim_{\ell \to \infty} \prod_{i=1}^{\ell} \mathbb{P}(X^{(i)}|Y^{(i)} = \Phi X^{(i)}; \Lambda) \tag{3}$$

$$= \arg \max_{\Phi, \Lambda} \mathbb{E}_{\mathbb{P}(X)}[\log(\mathbb{P}(X|Y = \Phi X; \Lambda))].$$

In practice and since we do not know the true underlying probability distribution of $\mathbb{P}(X|\widehat{X})$, we maximize a parametric distribution $q(X|\widehat{X})$ instead. In this case, in the asymptotic setting we can write

$$\Phi', \Lambda' = \arg \max_{\Phi, \Lambda} \lim_{\ell \to \infty} \prod_{i=1}^{\ell} q(X^{(i)}|Y^{(i)} = \Phi X^{(i)}; \Lambda) \tag{4}$$

$$= \arg \max_{\Phi, \Lambda} \mathbb{E}_{\mathbb{P}(X)}[\log(q(X|Y = \Phi X; \Lambda))].$$

Therefore, since Kullback–Leibler divergence is bounded above zero we have

$$\mathbb{E}_{\mathbb{P}(X)}[\log(q(X|Y = \Phi X; \Lambda))] \leq \mathbb{E}_{\mathbb{P}(X)}[\log(\mathbb{P}(X|Y = \Phi X; \Lambda))],$$

meaning that learning a parametric distribution for reconstructing $X$ from $Y$ is equivalent to maximizing a lower-bound of true conditional entropy and accordingly, mutual information between the input signal $X$ and undersampled measurements $Y$. Hence, although we are not maximizing the mutual information between $X$ and $Y$, we are maximizing a lower-bound of it through learning $\Phi$ and $\Lambda$. If we assume $X = \widehat{X} + \epsilon$, where $\epsilon$ and has an isotropic Gaussian distribution, then, since $q(X|\widehat{X} = \widehat{\mathbf{x}}) = \mathcal{N}(\widehat{\mathbf{x}}, \lambda \mathbb{I})$, the above maximization may be performed by minimizing the mean squared error (MSE).

Table 1: The isometry constant values of DeepSSRR low-dimensional embedding matrix $\Phi$ with different numbers of layers and filter sizes ($M = 256$).

| Num. Layers | 1 | 2 | 3 | 4 |
|---|---|---|---|---|
| $3 \times 3$ Filters | 0.289 | 0.237 | 0.186 | 0.174 |
| $5 \times 5$ Filters | 0.280 | 0.199 | 0.175 | 0.165 |

## 2.2 Applications of Low-Dimensional Embedding

DeepSSRR is mainly designed for jointly sensing and recovering sparse signals for CS applications. However, we can specifically train the sensing part of DeepSSRR (without using the recovery part) for several important dimensionality reduction tasks. The sensing part of DeepSSRR (i.e., the encoder or matrix $\Phi$) is a linear low-dimensional embedding that we can apply it to learn a mapping from a subset of $\mathbb{R}^N$ to $\mathbb{R}^M$ ($M < N$) that is a near-isometry, i.e., a mapping that nearly preserves all inter-point distances. This problem has a range of applications, from approximate nearest neighbor search to the design of sensing matrices for CS. Recall that, for a set $\mathcal{Q} \subset \mathbb{R}^N$ and $\epsilon > 0$, the (linear or nonlinear) mapping $\Phi : \mathcal{Q} \to \mathbb{R}^M$ is an $\epsilon$-isometry w.r.t the $\ell_2$-norm if for every $\mathbf{x}$ and $\mathbf{x}'$ in $\mathcal{Q}$ we have $(1 - \epsilon)\|X - X'\|_2 \leq \|\Phi(X) - \Phi(X')\|_2 \leq (1 + \epsilon)\|X - X'\|_2$.

Algorithm 1 demonstrates the use of the low-dimensional embedding matrix $\Phi$ of DeepSSRR to construct a near-isometric embedding. We achieve this by penalizing the maximum deviation from isometry in several batches of data that are created by permuting the original training data in every training epoch. In Section 3 we will show how our proposed algorithm works compared to competing methods.

## 3 Experimental Results

We now illustrate the performance of DeepSSRR against competing methods in several problems.

### 3.1 Linear Low-Dimensional Embedding

We first study the quality of the linear embeddings produced by DeepSSRR and its comparison with two other linear algorithms – NuMax (Hegde et al., 2015) and random Gaussian projections. To show the price of linearity, we also pit these against the nonlinear version of DeepSSRR and a DCN (eight nonlinear convolutional layers + a max-pooling layer). We use the grayscale version of CIFAR-10 dataset (50,000 training + 10,000 test $32 \times 32$ images). We train DeepSSRR and DCN according to Algorithm 1 by using filters of size $5 \times 5$. For DeepSSRR, depending on the size of the embedding we use five to seven layers to learn $\Phi$ in Algorithm 1.

Figure 3(a) shows the size of embedding $M$ as a function of the isometry constant $\epsilon$ for different methods. For the random Gaussian projections we have considered 100 trials and the horizontal error bars represent the deviation from average value. As we can see, the nonlinear version of DeepSSRR low-dimensional embedding outperforms almost all the other methods by achieving a given isometry constant with fewer measurements. The only exception is when $\epsilon > 0.6$ (i.e., a regime where we are not demanding a good isometry), where the DCN outperforms the nonlinear version of DeepSSRR; though, with more number of parameters.

**Effect of Number of Layers.** A convolutional layer is equivalent to the product of a circulant matrix and the vectorized input. The number of nonzero elements in a circulant matrix depends on the size of the convolution filter. As the number of such layers grows, so does the number of nonzero elements in the final embedding matrix. There are lower bounds (Nelson & Nguyen, 2013) on the number of nonzero elements in a matrix to ensure it is near-isometric. Table 1 shows the isometry constant value $\epsilon$ of DeepSSRR's low-dimensional embedding with different number of layers and different filter sizes. As we can see, $\epsilon$ gets smaller as the final embedding matrix has more nonzero elements (more layers, larger filters).

**Approximate Nearest Neighbors.** Finding the closest $k$ points to a given query datapoint is challenging for high-dimensional datasets. One solution is to create a near-isometric embedding that

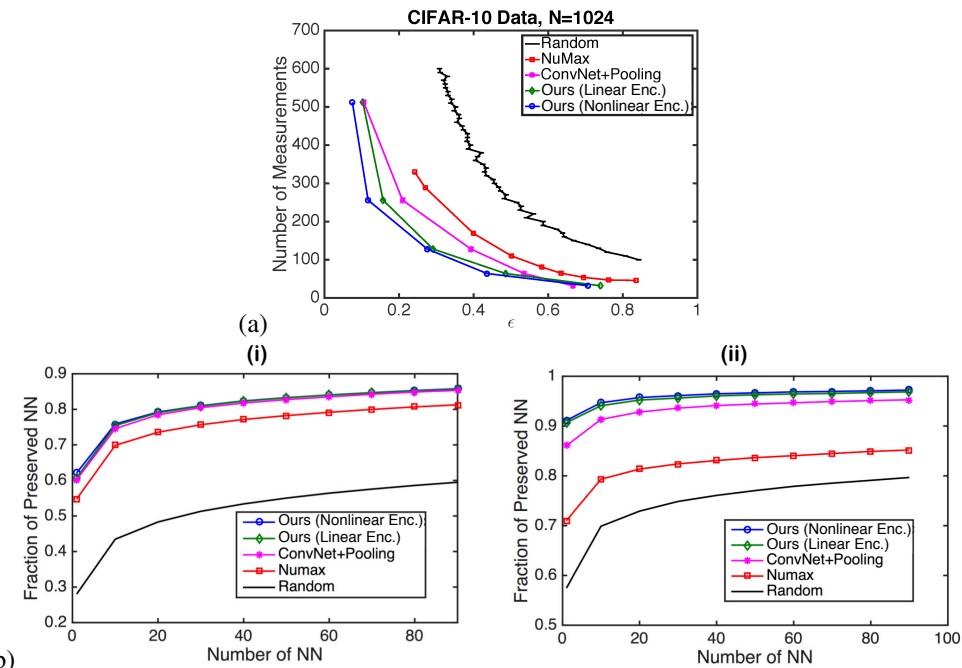

Figure 3: (a) Embedding size $M$ vs. empirical isometry constant $\epsilon$ for CIFAR-10 dataset. DeepSSRR significantly outperforms other methods for a wide range of $\epsilon$ values. (b) Fraction of $k$-nearest neighbors that are preserved in an $M$-dimensional embedding compared to the $N$-dimensional data for CIFAR-10 images. For NuMax and random embedding $M = 65$ in (i) and $M = 289$ in (ii). For deep networks (DeepSSRR and DCN) $M = 64$ in (i) and $M = 256$ in (ii).

maps datapoints from $\mathbb{R}^N$ to $\mathbb{R}^M$ ($M < N$) and solving the approximate nearest neighbors (ANN) problem in the embedded space. Fig. 3(b) compares the performance of different methods in the ANN problem. It shows the fraction of $k$-nearest neighbors that are retained when embedding datapoints in a low-dimensional space. We have considered two separate embedding problems: First $M = 65$ for random embedding and NuMax and $M = 64$ for DCN and DeepSSRR's low-dimensional embedding. Second, $M = 289$ for random embedding and NuMax and $M = 256$ for DCN and DeepSSRR's low-dimensional embedding. Since the size of the embedding for DCN and DeepSSRR's low-dimensional embedding is smaller in both settings, they have a more challenging task to find the nearest neighbors. As shown in Figure 3(b) DeepSSRR's low-dimensional embedding outperforms other approaches.

## 3.2 SIGNAL RECOVERY

We divide the discussion of this section into two parts. In the first part, we study the performance of DeepSSRR in the sparse signal recovery problem. The discussion of this part along with experimental results showing the effect of learning a sparse representation and parallelization on different criteria (e.g. phase transition, recovery accuracy and speed) are provided in **Appendix A**. In the second part that we provide in the following, we study the performance of DeepSSRR for the compressive image recovery problem.

**Compressive Image Recovery.** In this part, we study the compressive image recovery problem by comparing DeepSSRR with two state-of-the-art algorithms DAMP (Metzler et al., 2016) and LDAMP (Metzler et al., 2017). Both DAMP and LDAMP use random Gaussian $\Phi$ while DeepSSRR learns a $\Phi$. Here we run DAMP for 10 iterations and use a BM3D denoiser at every iteration. We also run LDAMP for 10 layers and use a 20-layer DCN in every layer as a denoiser. For DeepSSRR, we use 7 layers to learn the $\Phi$ and 7 layers to learn the $f_\Lambda(.)$. DeepSSRR is trained with an initial learning rate of 0.001 that is changed to 0.0001 when the validation error stops decreasing. For training, we have used batches of 128 images of size $64 \times 64$ from ImageNet (Russakovsky et al., 2015). Our training and validation sets include 10,000 and 500 images, respectively. Figure 4(a)

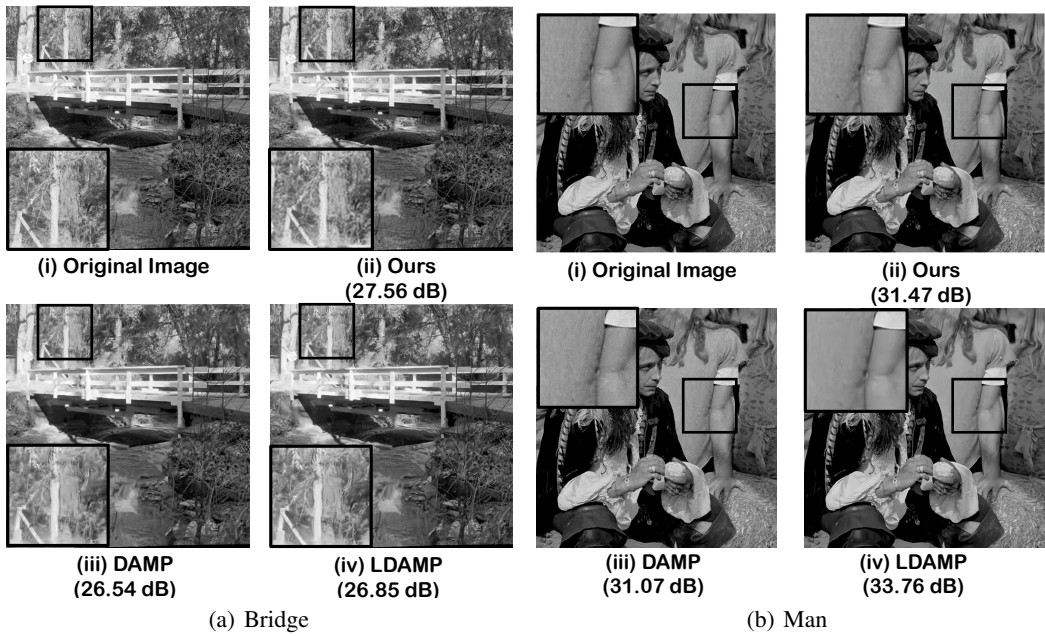

|  |  |  |  |
| :---: | :---: | :---: | :---: |
| **(i) Original Image** | **(ii) Ours**
**(27.56 dB)** | **(i) Original Image** | **(ii) Ours**
**(31.47 dB)** |
| **(iii) DAMP**
**(26.54 dB)** | **(iv) LDAMP**
**(26.85 dB)** | **(iii) DAMP**
**(31.07 dB)** | **(iv) LDAMP**
**(33.76 dB)** |
| (a) Bridge | | (b) Man | |

Figure 4: Reconstructions of $512 \times 512$ test images sampled at a rate of $\frac{M}{N} = 0.25$. DeepSSRR does a better job in recovering fine textures as compared to DAMP and LDAMP.

shows the reconstructions of the Bridge image ($\frac{M}{N} = 0.25$). DeepSSRR outperforms both DAMP and LDAMP in terms of accuracy (PSNR[1]). In particular, DeepSSRR does a better job at recovering the fine textures inside an image. Figure 4(b) presents the reconstructions of the Man image ($\frac{M}{N} = 0.25$). Although in this example LDAMP outperforms DeepSSRR in terms of PSNR, DeepSSRR does a better job at recovering fine textures. In this example LDAMP contains 10 unrolled iterations where each iteration contains a 20-layer DCN. In other words, LDAMP uses 200 convolutional layers in total. On the other hand, DeepSSRR uses only 7 convolutional layers to recover the Man image which is significantly smaller compared to LDAMP's number of layers. Iterative recovery algorithms and their unrolled versions such as DAMP and LDAMP typically involve a matrix vector multiplication in every iteration or layer, and hence their computational complexity is $\mathcal{O}(MN)$. In DeepSSRR, the length of feature maps in every convolutional layer is equal to the size of embedding $M$. Therefore, computing the output of typical middle layers will cost $\mathcal{O}(M)$ that is significantly cheaper than the one for iterative or unrolled methods such as DAMP and LDAMP.

**Effect of the Number of Layers.** Our experiments indicate that having more number of layers does not necessarily result in a better signal recovery performance. This phenomenon is also observed in Dong et al. (2016) for the image super-resolution problem. The reason for this problem is the increased non-convexity and non-smoothness of loss function as we add more layers. One way to mitigate this problem is to add skip connections between layers. As shown in Li et al. (2017), skip connections smooth the loss surface of deep networks and make the optimization problem simpler.

## 4 CONCLUSIONS

In this paper we introduced DeepSSRR, a framework that can learn both near-optimal sensing schemes, and fast signal recovery procedures. Our findings set the stage for several directions for future exploration including the incorporation of adversarial training and its comparison with other methods (Bora et al., 2017; Dumoulin et al., 2016; Donahue et al., 2016). Furthermore, a major question arising from our work is quantifying the generalizability of a DeepSSRR-learned model based on the richness of training data. We leave the exploration of this for future research.

---

[1]PSNR $= 10 \log_{10}(1/\text{MSE})$ when the pixel range is 0 to 1.

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

APPENDIX

## A    SPARSE SIGNAL RECOVERY

In this section we study the problem of sparse signal recovery by comparing DeepSSRR to another DCN called DeepInverse (Mousavi & Baraniuk, 2017) and to the LASSO (Tibshirani, 1996) $\ell_1$-solver implemented using the coordinate descent algorithm of Friedman et al. (2010). We assume that the optimal regularization parameter of the LASSO is given by an oracle in order to obtain its best possible performance. Also, both training and test sets are wavelet-sparsified versions of 1D signals of size $N = 512$ extracted from rows of CIFAR-10 images and contain 100,000 and 20,000 signals, respectively. While DeepSSRR learns how to take undersampled measurements of data through its low-dimensional embedding $\Phi$, DeepInverse uses random undersampling (i.e., a random $\Phi$). DeepSSRR in this section has 3 layers for learning $\Phi$ and 3 layers for learning $f_\Lambda(.)$ with filter size $25 \times 1$ while DeepInverse has five layers for learning the inverse mapping with filter size $125 \times 1$.

Figure 5(a) shows the $\ell_1$ phase transition plot (Donoho et al., 2009). This plot associates each grid point to an ordered pair $(\delta, \rho) \in [0, 1]^2$, where $\delta = \frac{M}{N}$ denotes the undersampling ratio and $\rho = \frac{K}{M}$ denotes the normalized sparsity level. Each grid point $(\delta, \rho)$ represents the probability of an algorithm's success in signal recovery for that particular problem configuration. As the name suggests, there is a sharp phase transition between values of $(\delta, \rho)$ where recovery fails with high probability to when it succeeds with high probability. In Figure 5(a), the blue curve is the $\ell_1$ phase transition curve. The circular points denote the problem instances on which we study the performance of DeepInverse and the LASSO. The square points denote the problem instances on which we have trained and tested DeepSSRR. By design, all these problem instances are on the "failure" side of the $\ell_1$ phase transition. For DeepSSRR (square points), we have made recovery problems harder by reducing $\delta$ and increasing $\rho$. The arrows between the square points and circular points in Figure 5(a) denote correspondence between problem instances in DeepSSRR and DeepInverse. Table 2 shows the average normalized MSE (NMSE) for the test signals. While DeepSSRR recovers the same signals from fewer measurements, it outperforms DeepInverse and the LASSO. DeepSSRR outperforms DeepInverse while having significantly fewer number of parameters (less than 70,000 vs. approximately 200,000 parameters). This is mainly due to the fact that DeepSSRR learns $\Phi$ instead of using a random $\Phi$ as is the case in DeepInverse and conventional CS.

Fig. 5(b) compares training speed of DeepInverse and DeepSSRR. It shows the MSE of recovering test signals by DeepInverse and DeepSSRR in different training epochs. While training and test sets are the same, the configuration for DeepInverse (and LASSO) is $(\delta, \rho) = (0.7, 0.72)$ and for DeepSSRR is $(\delta, \rho) = (0.5, 1.003)$ which means we have given DeepSSRR a more challenging problem. As shown in Fig. 5(b), due to the extra parallelization scheme (i.e., rearrangement layer) convergence is significantly faster for DeepSSRR compared to DeepInverse. DeepSSRR outperforms the LASSO after only 4 training epochs while DeepInverse takes 138 epochs. This fast convergence has two major reasons: First, DeepSSRR has fewer number of parameters to learn. Second, DeepSSRR learns adaptive measurements (i.e., low-dimensional embedding) instead of using random measurements (i.e., random embedding).

Fig. 5(c) compares the probability of successful recovery by DeepSSRR and LASSO as measured by 20,000 Monte Carlo test samples. For each undersampling ratio $\delta$ and for the $j$-th Monte Carlo sample, we define the success variable $\varphi_{\delta,j} = \mathbb{I}\left(\frac{\|\hat{X}^{(j)} - X^{(j)}\|_2^2}{\|X^{(j)}\|_2^2} \leq 0.01\right)$, where $X^{(j)}$ is the $j$-th sample, $\hat{X}^{(j)}$ is the recovered signal from measurements of $j$-th sample, and $\mathbb{I}(.)$ is the indicator function. We denote empirical successful recovery probability by $P_\delta = \frac{1}{q}\sum_{j=1}^{q} \varphi_{\delta,j}$. In Fig. 5(c), our test samples are $k$-sparse where $k = 34$, and we have considered three different configurations: $M = 64, 128, 256$ that correspond to above, on, and below the $\ell_1$ phase transition, respectively. As we can see in Fig. 5(c), DeepSSRR significantly outperforms LASSO when the problem configuration lies above (failure phase) or on the $\ell_1$ phase transition and LASSO slightly outperforms when the problem configuration lies below the $\ell_1$ phase transition (success phase). For a setting below the $\ell_1$ phase transition, we expect $\ell_1$ minimization to behave the same as $\ell_0$ minimization. However, DeepSSRR should learn a transformation for transforming measurements back to the original signals. Furthermore, Fig. 5(c) shows the price we pay for using a linear low-dimensional embedding $\Phi X$

Table 2: Average NMSE of test set signals recovered with different configurations and methods.

| $(\delta, \rho)$ | LASSO | DeepInverse | $(\delta, \rho)$ | Ours |
|---|---|---|---|---|
| (0.3,0.42) | 0.0466 | 0.0140 | (0.25,1.12) | **0.0137** |
| (0.7,0.72) | 0.0164 | 0.0104 | (0.5,1.003) | **0.0057** |

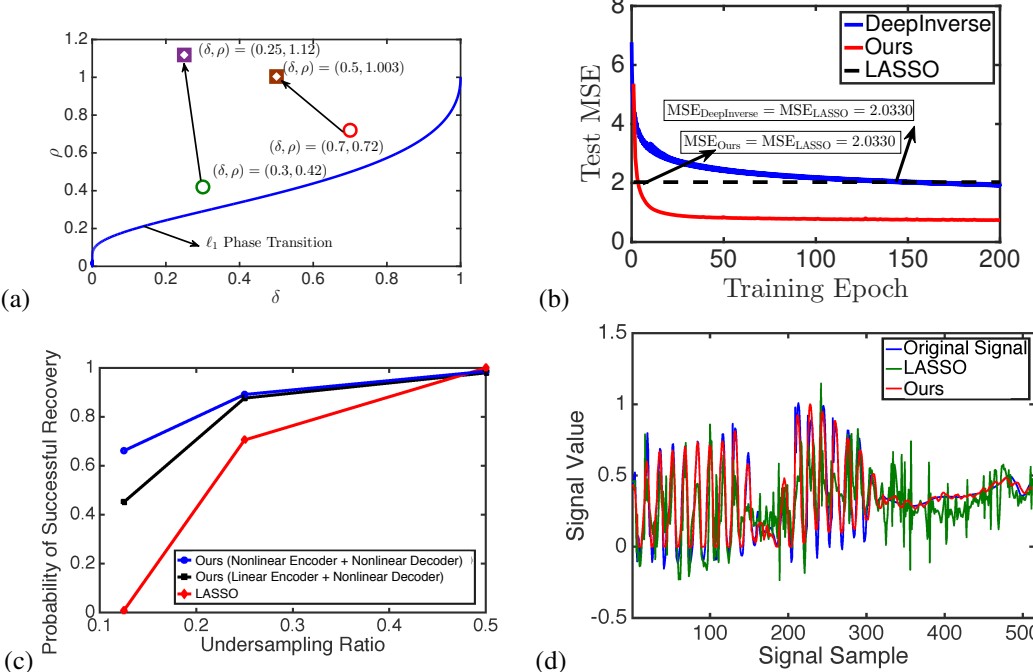

(a)  (b)  (c)  (d)

Figure 5: (a) The blue curve is the $\ell_1$ phase transition. The circular and square points denote problem instances for DeepInverse and DeepSSRR, respectively. Arrows show different configurations for the same set of signals. (b) Test MSE of DeepInverse and DeepSSRR during different training epochs. DeepSSRR outperforms LASSO after significantly fewer training epochs compared to DeepInverse. (c) Average probability of successful recovery for three different configurations: below, above, and on the $\ell_1$ phase transition. (d) Recovery example by DeepSSRR and LASSO (with optimal regularization parameter). The $(\delta, \rho)$ lies on the failure side of $\ell_1$ phase transition.

instead of a nonlinear one $\Phi(X)$. The main message of Figure 5(c) is that by using DeepSSRR we can have a significantly better phase transition compared to $\ell_1$-minimization.

Fig. 5(d) shows examples of signal recovery using DeepSSRR and LASSO. The test signal is $k$-sparse where $k = 64$. DeepSSRR and LASSO recover this signal from $M = 64$ and $M = 154$ measurements, respectively. DeepSSRR has solved a more challenging recovery problem better than the LASSO with optimal regularization parameter.

# B  COMPRESSIVE IMAGE RECOVERY

In this section we study another example of the compressive image recovery problem. The settings we have used in here is exactly the same as Section 3.2. Fig. 6 shows the reconstruction of the mandrill image ($\frac{M}{N} = 0.25$). Fig. 6(a) shows the reconstruction of whole face and Fig. 6(b) shows the reconstruction of nose and cheeks. As we can see, although LDAMP slightly outperforms our method in Fig. 6(a), our method does a significantly better job in recovering the texture of nose and cheeks in Fig. 6(b). Not only our method outperforms LDAMP by 0.9 dB, but also it has a better visual quality and fewer artifacts (e.g. less over-smoothing).

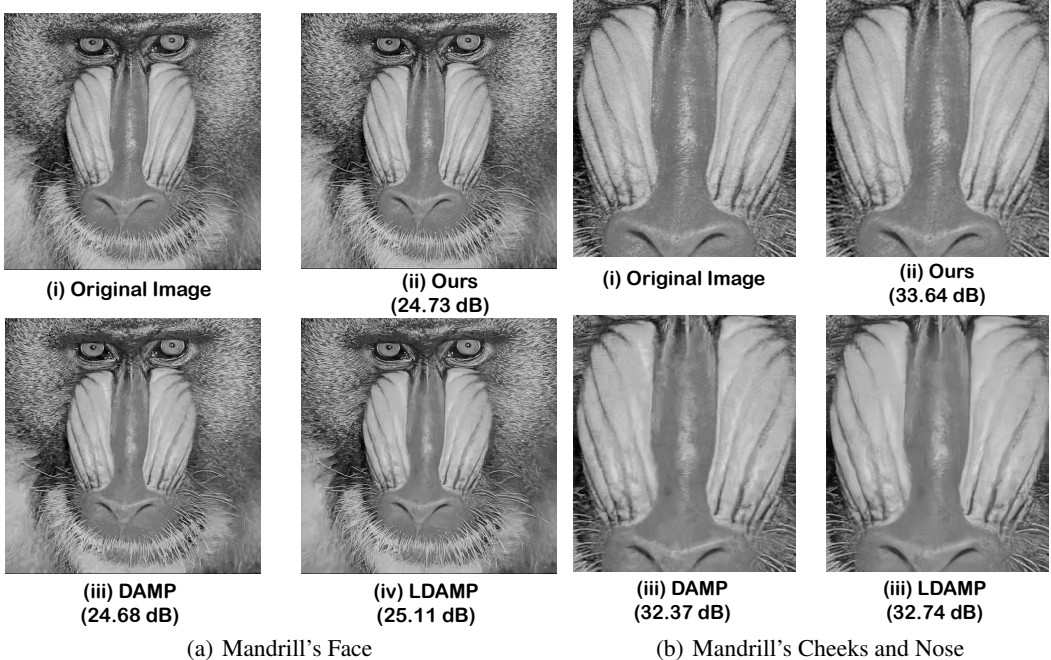



**(i) Original Image**      **(ii) Ours**
(24.73 dB)      **(i) Original Image**     **(ii) Ours**
(33.64 dB)

**(iii) DAMP**
(24.68 dB)     **(iv) LDAMP**
(25.11 dB)     **(iii) DAMP**
(32.37 dB)     **(iii) LDAMP**
(32.74 dB)

(a) Mandrill's Face      (b) Mandrill's Cheeks and Nose



Figure 6: (a) Reconstruction of the $512 \times 512$ mandrill image sampled at the rate of $\frac{M}{N} = 0.25$. (b) Zoomed in reconstruction of mandrill's nose and cheeks.

| Algorithm | DAMP | LDAMP | Ours |
|---|---|---|---|
| **Running Time** | 77.24 sec | 1.62 sec | **0.41 sec** |

Table 3: Running time of different algorithms for the reconstruction of a $512 \times 512$ image where undersampling ratio, i.e. $\frac{M}{N}$ is 0.25.

## C    RUNNING TIME

In this section we compare the running time of different algorithms. We consider the reconstruction of a $512 \times 512$ image with an undersampling ratio of $\frac{M}{N} = 0.25$. Table 3 shows the comparison between different algorithms. We should note that authors in Metzler et al. (2017) have used coded diffraction pattern in DAMP and LDAMP which simplifies the computational complexity of vector-matrix multiplications in DAMP and LDAMP to $\mathcal{O}(N \log(N))$ instead of $\mathcal{O}(MN)$. In addition, we should note that LDAMP uses filters of size $3 \times 3$ in its convolutional layers while we use filters of size $5 \times 5$ in the convolutional layers of our architecture. Table 3 shows that our method is almost 4 times faster than the LDAMP method.

