# OpenReview forum: "A Data-Driven and Distributed Approach to Sparse Signal Representation and Recovery"
_ICLR.cc/2019/Conference_

### Official Review · AnonReviewer3 · 2018-10-26
**A data-driven and distributed approach to sparse signal representation and recovery**

**Rating:** 6
**Confidence:** 3

**Review:**

Authors case the problem of finding informative measurements by using a maximum likelihood formulation and show how a data-driven dimensionality reduction protocol is built for sensing signals using convolutional architectures. A novel parallelization scheme is discussed and analyzed for speeding up the signal recovery process.

Previous works have been proposed to jointly learn the signal sensing and reconstruction algorithm using convolutional networks. Authors do not consider them as the baseline methods due to the fact that the blocky reconstruction approach is unrealistic such as MRI. However, there is no empirical result to support his conclusion.  In addition, the comparisons to these methods can further convince the readers about the advantage of the proposed method.

It is not clear where the maximum deviation from isometry in Algorithm 1 is discussed since the MSE is used as a loss function.

Authors provided theoretical insights for the proposed algorithm. It indicates that the lower-bound of the mutual information is maximized and minimizing the mean squared error is a special case, but it is unclear why this can provide theoretical guarantee for the proposed method. More details are good for the connections between the theory and the proposed algorithm.

One of the contributions in this paper is the speed, so the results on the speed should be put in the main paper.

---

> ### Author Response · Authors · 2018-11-27
> **Response to Comments**
>
>
> Comment: Previous works have been proposed to ...
>
> Response: As discussed in "Compressed Sensing MRI" (available at https://ieeexplore.ieee.org/abstract/document/4472246), in CS MRI, the measurement matrix is basically a subsampled Fourier matrix. On the other hand, in algorithms that reconstruct signals block-by-block (similar to Kulkarni et al. 2016 and Shi et al. 2017), the measurement matrix has a block-diagonal structure. Now since the entries of a subsampled Fourier matrix cover a range of frequencies, it cannot have a block-diagonal structure. This means that methods that reconstruct signals block-by-block cannot use a subsampled Fourier matrix as their measurement matrix. Hence, these methods are not suitable for the MRI application.
>
> Also, please note that the reconstruction we obtain from algorithms that reconstruct signals block-by-block have clear blocky artifacts. Because of the blocky artifacts, these algorithms typically include an additional denoiser to suppress the artifacts. In order to have a fair comparison between our approach and these previous works, we should also train a separate denoiser at the end of our architecture to improve its final reconstruction. However, this was beyond the scope of our work. Fortunately, our method achieves the state-of-the-art results in terms of compressive image recovery quality even without an extra denoiser.
>
> Comment: It is not clear where the maximum deviation from isometry in Algorithm 1 is discussed since the MSE is used as a loss function.
>
> Response: Please note that Algorithm 1 is related to the Section 2.2 titled  "Applications of Low-Dimensional Embedding". In this section, we have discussed how we can use "the encoder" part of our approach to build near-isometric embeddings. Therefore, the loss function that we have used in this section (which only uses the encoder for sensing) is the maximum deviation from isometry as you can see in Algorithm 1 (one line above the first 'end for'). However, when we use both encoder and decoder (for both sensing and recovery), we use MSE as the loss function.
>
> Comment: Authors provided theoretical insights for the proposed algorithm ...
>
> Response: Note that our theoretical insights are just that; in particular, we do not intend them to be interpreted as rigorous theoretical guarantees. Indeed establishing such guarantees is an exciting avenue for future work. With this in mind, we have argued why learning both Phi and Lambda (sensing and recovery) simultaneously, which is exactly what our approach does, is useful for compressive sensing (CS). We have characterized good measurements as the ones that give us back the original signals with the highest probability. We have shown that this problem is equivalent to maximizing the mutual information between the measurements and the original signals. Then we have argued that, since in practice our training data is limited and we do not know the exact distribution of data, we are not able to maximize this mutual information. Instead, we can assume a parametric distribution on the reconstruction error and show that jointly learning Phi and Lambda (i.e., sensing and recovery) gives us a lower-bound on the mutual information we wanted to maximize. Although we do not have rigorous theoretical guarantees for our approach, we have demonstrated that in practice it works very well.
>
> Comment: One of the contributions in this paper is the speed, so the results on the speed should be put in the main paper.
>
> Response: We have added a new section in the Appendix to discuss the computational benefits of our approach. As shown in Table 3 of our updated manuscript, our method is significantly faster than both DAMP and LDAMP methods.

---

### Official Review · AnonReviewer2 · 2018-10-29
**Interesting theory, shakey experiments**

**Rating:** 7
**Confidence:** 3

**Review:**

Quality & Clarity:
This is a nice paper with clear explanations and justifications. The experiments seem a little shakey.

Originality & Significance:
I'm personally not familiar enough to say the theoretical work is original, but it is presented as so. However it seems significant. The numerical results do not seem extremely significant, but to be fair I'm not familiar with state of the art nearest neighbor results ie Fig 3.

Pros:
I like that you don't take much for granted. E.g. you justify using convolutional net in 2.1, and answered multiple of my questions before I could type them (e.g. why didn't you include nonlinearities between convolutions, why bother with cascaded convolutions, and what you mean by near-optimal).

Cons:
The visual comparisons in Figure 4 are difficult to see. DLAMP appears to be over-smoothing but in general it's hard to compare to low-ish resolution noisy-looking textures. I strongly recommend using a test image with a clear texture to illustrate your point (eg the famous natural test image that has on the side a tablecloth with zig-zag lines)

The horizontal error bars are obfuscated by the lines between markers in Fig 3a.

I don't understand Fig 3a. You are varying M, which is on the Y-axis, and observing epsilon, on the X-axis?

Questions:
Can you state what is novel about the discussion in the "Theoretical Insights" subsection of 2.1? I guess this is described in your abstract as "we cast the problem ... by using a maximum likelihood protocol..." but your contribution could be made more explicit. For example "We show that by jointly optimizing phi and lambda (sensing and recovery), we are maximizing the lower bound of mutual information between reconstructions (X) and samples (Y)" (that is my understanding of the section)

Why don't you use the same M for all methods in the Figure 3 experiments? ie why did you use a different M for numax/random versus deepSSRR/DCN?

Why do you choose 20-layers for the denoiser? Seems deep...

The last part of the last sentence of the 2nd paragraph of section 3.1 should be a complete sentence "though, with more number of parameters". Does that mean that the DCN has more parameters than the DeepSSRR?

I am willing to change score based on the response

******************
Update after author response:
Thanks for the clear response and Figure 3, and nice paper. My score is updated.
PS: I still think that the (tiny) error bars are obfuscated because the line connecting them is the same thickness and color.

---

> ### Author Response · Authors · 2018-11-27
> **Response to Questions**
>
>
> Question: Can you state what is novel ...
>
> Response: The "Theoretical Insights" section basically describes why learning phi and lambda (sensing and recovery) simultaneously is useful for the CS problem. We characterize "good" measurements as the ones that give us back the original data with the highest probability. This problem is equivalent to maximizing the mutual information between the original data and the measurements. Since we do not know the true underlying distribution of data, we cannot maximize this mutual information. Instead, we assume a parametric distribution on the reconstruction error and show that jointly learning the sensing and recovery gives us a lower-bound on the mutual information we wish to maximize. In other words, instead of maximizing the true mutual information, we maximize a lower-bound of it.
>
> Question: Why don't you use the same M ...
>
> Response: In response to one of the previous comments, we described an important difference between NuMax and other approaches (in terms of the input/output of the algorithms). Because of this difference, we were not able to use the exact same M for all approaches. In addition, since calculating \epsilon is significantly cheaper for random embedding compared to deep learning-based approaches, we have used a fewer number of 'M's for the curves of learning-based approaches (i.e., DCN/DeepSSRR) compared to the curve of random embedding.
>
> Question: Why do you choose 20-layers for the denoiser? Seems deep...:
>
> Response: The LDAMP paper uses DnCNN, which is a 20-layer convolutional network. The reason for having 20 layers can be understood from the Table 1 of the DnCNN paper (https://arxiv.org/pdf/1608.03981.pdf), which tabulates the effective patch size for different denoisers. Considering the fact that DnCNN uses convolutional layers with 3x3 filters, the authors have chosen 20 layers in order to have a receptive field (which is correlated to the effective patch size) similar to other denoisers. For a more detailed argument, please refer to Section III.A of DnCNN paper (https://arxiv.org/pdf/1608.03981.pdf).
>
> Question: The last part of the last sentence of the 2nd paragraph ...
>
> Response: Yes, it means that the DCN has more parameters compared to DeepSSRR. As we have mentioned in the 1st paragraph of Section 3.1, the DCN has 8 convolutional layers, while DeepSSRR has 5 to 7 convolutional layers, depending on the size of embedding.

---

> ### Author Response · Authors · 2018-11-27
> **Response to Cons mentioned by the reviewer**
>
>
> Comment: The visual comparisons in Figure 4 ...
>
> Response: We have added a new visual comparison (Figure 6) in our updated manuscript that presents the reconstruction of the 512x512 Mandrill test image with sampling ratio= 0.25. In this case, LDAMP slightly outperforms our algorithm. However, in order to compare the reconstruction of textures, we have explicitly compared the reconstruction of Mandrill's nose and cheeks. Figure 6(b) shows that, in this case, our algorithm outperforms LDAMP by 0.9dB and has a better visual quality and fewer artifacts (e.g. less over-smoothing).
>
> Comment: The horizontal error bars ...
>
> Response: Please note that we have used horizontal error bars only for the random embedding, which does not have any marker. If you zoom in on the plot, these horizontal error bars are well concentrated around their mean values.
>
> Comment: I don't understand Fig 3a ...
>
> Response: First of all, there is an important difference between NuMax and the other algorithms in Figure 3a. In Algorithm 1 of the NuMax paper (http://home.engineering.iastate.edu/~chinmay/files/papers/numax_tsp.pdf), the parameter \epsilon (which is called \delta in that paper) is an input to the algorithm. Given a value for \epsilon, NuMax determines the appropriate dimension of the embedding (i.e., M). However, for other approaches (random/DeepSSRR/DCN) we do not give an \epsilon to the algorithm. Instead, we pick an embedding size (i.e., M), construct an embedding of that size, and then measure the \epsilon. In other words, for NuMax, \epsilon is the input and M is the output while for other methods, M is the input and \epsilon is the output. In spite of this difference, the visualization in Figure 3a lets us compare different methods and understand which one gives us a better isometry constant.

---

### Official Review · AnonReviewer1 · 2018-11-05
**Review: An interesting approach to data-driven compressed sensing**

**Rating:** 8
**Confidence:** 4

**Review:**


This paper proposes a (CNNs) architecture for encoding and decoding images for compressed sensing.
In standard compressed sensing (CS), encoding usually is linear and corresponds to multiplying by a fat matrix that is iid gaussian. The decoding is performed with a recovery algorithm that tries to explain the linear measurements but also promotes sparsity. Standard decoding algorithms include Lasso (i.e. l1 regularization and a MSE constraint)
or iterative algorithms that promote sparsity by construction.

This paper instead proposes a joint framework to learn a measurement matrix Phi and a decoder which is another CNN in a data-driven way. The proposed architecture is novel and interesting.

I particularly liked the theoretical motivation of the used MSE loss by maximizing mutual information.

The use of parallel convolutions is also neat and can significantly accelerate inference, which can be useful for some applications.

The empirical performance is very good and matches or outperforms previous state of the art reconstruction algorithms D-AMP and Learned D-Amp.

On comparisons with prior/concurrent work: The paper is essentially a CNN autoencoder architecture but specifically designed for compressed sensing problems.
There is vast literature on CNN autoencoders including (Jiang 2017 and Shi 2017) paper cited by the authors. I think it is fine to not compare against those since they divide the images into small blocks and hence have are a fundamentally different approach. This is fine even if block-reconstruction methods outperform this paper, in my opinion: new ideas should be allowed to be published even if they do not beat SOTA, as long as they have clearly novel ideas. It is important however to discuss these differences as the authors have done in page 2.

Specific comments:

1. It would be interesting to see a comparison to D-Amp and LDAmp for different number of measurements or for different SNRs (i.e. when y = Phi x+ noise ). I suspect each method will be better for a different regime?

2. The paper: `The Sparse Recovery Autoencoder' (SRA) by Wu et al. https://arxiv.org/abs/1806.10175
is related in that it learns both the sensing matrix and a decoder and is also focused on compressed sensing, but for non-image data. The authors should discuss the differences in architecture and training.

3. Building on the SRA paper, it is possible that the learned Phi matrix is used but then reconstruction is done with l1-minimization. How does that perform for the matrices learned with DeepSSRR?

4. Why is Figure 1 going from right to left?

---

> ### Author Response · Authors · 2018-11-27
> **Response to Comments 3 and 4**
>
> 3- We refer the reviewer to Section 2.2 of our submission ("Applications of Low-Dimensional Embedding"). In this section and in Algorithm 1, we discuss how we can learn near-isometric embeddings using our approach. One of the main applications of near-isometric embeddings is designing compressive sensing (CS) measurement matrices. In CS language, learning a near-isometric embedding is equivalent to learning a measurement matrix that satisfies the so-called restricted isometry property (RIP). RIP is a *sufficient* condition for compressive sensing. This means that the matrices we learn with Algorithm 1 can be used along with L1 minimization for CS.
>
> For a comparison of our approach with previous work, we refer the reviewer to Figure 3(a) in our submission and also Figure 8 in the NuMax paper we cite (available at http://home.engineering.iastate.edu/~chinmay/files/papers/numax_tsp.pdf). Figure 8 of the NuMax paper compares the CS recovery performance of NuMax vs. random Gaussian projections and shows that NuMax outperforms random projections in terms of MSE for different measurement ranges and SNRs. This success is mainly explained by Figure 3 of the NuMax paper, which shows that the matrices built by the NuMax algorithm have a better isometry constant than random matrices. With this in mind, we now refer the reviewer to Figure 3(a) in our manuscript, where we have shown that the isometry constant of our method is even better than NuMax. This means that, if our approach is used with L1 reconstruction, then the result will be better than using either random matrices or NuMax matrices. Therefore, the answer to the reviewer's question is "yes". We can basically use matrices learned with our approach along with L1 reconstruction, and the result will beat both random projections and NuMax embeddings.
>
> 4- We used a right to left ordering in Figure 1, because we wanted to include the vector-matrix multiplications denoted as 'parallel convolutions' in this figure.

---

> ### Author Response · Authors · 2018-11-27
> **Response to Comments 1 and 2**
>
> Re your specific comments:
>
> 1- It is indeed possible to compare learning-based approaches to compressive sensing (like our work in this manuscript) vs. model-based approaches (like AMP, DAMP). We refer the reviewer to Figure 5(c) in our paper and also Figure 3 of the paper "A Learning Approach to Compressed Sensing," http://cs231n.stanford.edu/reports/2017/pdfs/8.pdf. Figure 5 of our paper compares the performance of our learning-based approach vs. the LASSO L1 solver; Figure 3 of the aforementioned paper compares the performance of other learning-based approaches such as CNN and VAE with AMP. Both figures show that i) when the undersampling ratio (i.e. m/n) is small, learning-based approaches (like our work) can outperform model-based approaches (such as AMP or DAMP); ii) when the undersampling ratio is large enough, model-based approaches start to outperform learning-based approaches.
>
> Intuitively, when the undersampling ratio is large enough, model-based approaches can extract sufficient information from measurements to reconstruct signals accurately enough and even better than learning-based approaches.
> Moreover, model-based algorithms like AMP/DAMP have the knowledge of the measurement matrix and this is another factor helping them to be better than learning-based approaches in high undersampling ratio regime.
>
> Regarding different SNRs, we refer the reviewer to Table 3 of [arXiv:1701.03891] which we have also cited in our submission. In that table, the authors compare the robustness of recovery based on CNNs and DAMP. As they have shown, CNNs are more robust to noise. In general, learning-based approaches can utilize data to more effectively suppress measurement noise.
>
> Finally, we note that the LDAMP approach we have cited in our paper is very similar to DAMP except that, instead of using a BM3D denoiser, LDAMP uses a CNN denoiser. The rest of the architecture is not learned and hence, is similar to AMP/DAMP. Therefore, one can expect that LDAMP's behaviour would be similar to DAMP except for the fact that it has a better denoiser.
>
> 2- We have added a reference to the SRA paper in our revised paper plus added a short discussion of the differences with our approach. Like our approach, the SRA architecture is also an autoencoder. In SRA, the encoder can be considered to be a fully connected layer while in our work the encoder has a convolutional structure and is basically a circulant matrix. For large problems, learning a fully connected layer (as in the SRA encoder) is significantly more challenging than learning one/several convolutional layers (as in our encoder). In SRA, the decoder is a T-step projected subgradient. In our work, the decoder consists of several convolutional layers plus a rearrangement layer. The optimization in SRA is solely over the measurement matrix and T (which is the number of layers in the decoder) scalar values that could be considered as learning rates at every layer of the decoder. However, in our work, the optimization is over the convolution weights and biases that we have across the different layers of our encoder and decoder. The authors of SRA have shown results mainly on synthetic datasets whereas we have presented results on real images.

---

> > ### Comment · AnonReviewer1 · 2018-12-09
> > **post rebuttal**
> >
> > I think the authors have addressed all my comments and I recommend acceptance.

---

### Meta-Review · Area_Chair1 · 2018-12-14
**Trainable Image Compressed Sensing with solid empirical results**

**Confidence:** 4
**Recommendation:** Accept (Poster)

**Metareview:**

This paper studies deep convolutional architectures to perform compressive sensing of natural images, demonstrating improved empirical performance with an efficient pipeline.
Reviewers reached a consensus that this is an interesting contribution that advances data-driven methods for compressed sensing, despite some doubts about the experimental setup and the scope of the theoretical insights. We thus recommend acceptance as poster.